# FEC: Fast Euclidean Clustering for Point Cloud Segmentation

**Yu Cao** [1,†], **Yancheng Wang** [2,†], **Yifei Xue** [3] , **Huiqing Zhang** [1] and **Yizhen Lao** [1,*]

1 College of Computer Science and Electronic Engineering, Hunan University, Lushan Road (S), Yuelu District, Changsha 410012, China
2 Alibaba Group, Hangzhou 311121, China
3 Jiangxi Provincial Natural Resources Cause Development Center, Nanchang 330000, China
* Correspondence: yizhenlao@hnu.edu.cn
† These authors contributed equally to this work.

**Abstract:** Segmentation from point cloud data is essential in many applications, such as remote sensing, mobile robots, or autonomous cars. However, the point clouds captured by the 3D range sensor are commonly sparse and unstructured, challenging efficient segmentation. A fast solution for point cloud instance segmentation with small computational demands is lacking. To this end, we propose a novel fast Euclidean clustering (FEC) algorithm which applies a point-wise scheme over the cluster-wise scheme used in existing works. The proposed method avoids traversing every point constantly in each nested loop, which is time and memory-consuming. Our approach is conceptually simple, easy to implement (40 lines in C++), and achieves two orders of magnitudes faster against the classical segmentation methods while producing high-quality results.

**Keywords:** point cloud; instance segmentation; fast clustering

## 1. Introduction

The point cloud is a typical data structure deemed natural and intuitive to represent geometric space information in specific scenarios, ranging from a pen on a table to illustrating the world. People acquire this kind of data mainly by reconstruction from pairs of 2D images, or directly utilizing LiDAR devices (often mounted on backpacks, vehicles, drones, or aircraft). However, raw point cloud data are not always ready for use due to redundant information, uneven distribution, and object occlusion [1]. As shown in Figure 1 that point cloud segmentation is a common approach to tackle this issue by classifying point clouds with similar properties. Hence, point cloud segmentation becomes essential in 3D perception (computer vision, remote sensing) and 3D reconstruction (autonomous driving, virtual reality). For example, a robot should identify obstacles nearby to interact with and move around the scene [2]. Achieving this goal requires distinguishing between distinct semantic labels and various instances with the same semantic label. Thus, it is crucial to investigate the problem of point cloud segmentation.

### 1.1. Related Works

There is a substantial amount of work that targets acquiring a global point cloud and segmenting it off-line which can be classified into four main categories:

#### 1.1.1. Edge-Based Method

The most important procedure for Edge-based approaches is finding boundaries of each object section. Zucker and Hummel [3] extends edged detection methods from two-dimension (2D) to three-dimension (3D) level in geometrical way. This notion hypothesizes that the 3D scene can be divided by planes, and finding the edges is an intrinsic optimal way to get units. Then the segmentation problem simply to edge detection according to gradient, thresholds, morphological transform, filtering, template-matching [4–8]. Öztireli et al. [9]

introduce a parameter-free edge detection method with assistance of kernel regression. Rabbani et al. [10] summarise that this kind of method could be divided into two stages, namely border outlining and inner boundary grouping. However, the main drawback of edge-based methods is under-effective in noisy data [8].

Input unorganized point cloud                          Segmentation result

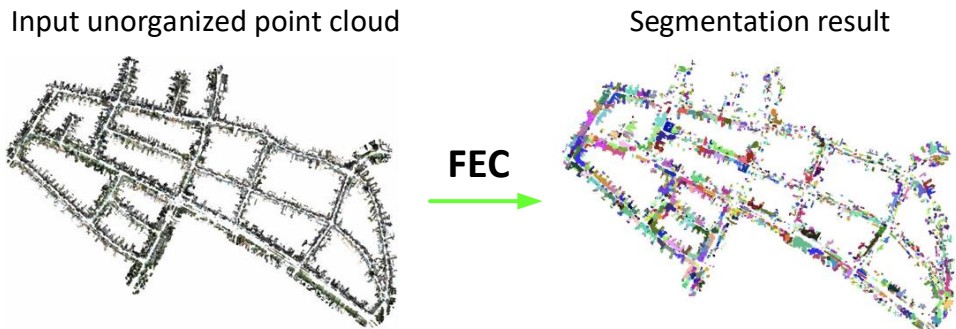

**FEC**

**Figure 1.** Unorganized point cloud instance segmentation results of proposed method FEC on one KITTI sequence [11] with 27 million points. FEC outperforms the existing segmentation approaches [10,12] with over 100× speedup rates.

1.1.2. Region Growing Based Method

Region growing (RG)-based methods focus more on extracting geometrical homogeneous surfaces than detecting boundaries. The assumption is that different parts of objects/scenes are visually distinguishable or separable. Generally speaking, these methods make point groups by checking each point (or derivatives, like voxels/supervoxels [13,14]) around one or more seeds by specific criteria. So researchers advocating RG-based method developed several criteria considering orientation [10,15], surface normal [10,16,17] , curvatures [18], et al. Li et al. [19] explore the possibility to apply RG algorithm to multiple conclusion cases (i.e., leaf phenotyping) and proves to be feasible. Typically, the conventional regional growth algorithm employs the RANSAC algorithm to generate seed spots. Considering internal features of point clouds, Habib and Lin [20] randomly select seeds and segment multi-class at once. Dimitrov and Golparvar-Fard [21] propose an RG-based method, which filters seed candidates by roughness measurement, for robust context-free segmentation of unordered point clouds based on geometrical continuities. However, the selection strategy of seed points is crucial cause if means to the computing speed and over- or under-segmentation [13]. In addition, these approaches are highly responsive to the imprecise calculation of normals and curvatures close to region borders [22].

1.1.3. Clustering Based Method

The clustering algorithms segment or simplify point cloud elements into categories based on their similarities or euclidean/non-euclidean distances. As a result, k-means [23], mean shift [24], DBSCAN [25], and euclidean cluster (EC) extraction [12] were employed on this task. The k-means clustering aims at grouping point data into $k$ divisions constrained by average distances [26]. Since point cloud data is often acquired unevenly, k-means is an ideal tool to remove redundant dense points [27]. Kong et al. [28] introduces a k-plane approach to categorize laser footprints that cannot be accurately classified using the standard k-means algorithm. DBSCAN assumes density distribution is the critical hint for deciding sample categories as dense regions, making it noise-resistant [25]. Even so, the DBSCAN method suffers from high memory capacity requirements for storing distance data between points. Chehata et al. [29] improve normal clustering methods to a hierarchical one, which is shown to be suitable for microrelief terrains. In order to compress point clouds without distortion and losing information, Sun et al. [30] demonstrate a 3D range-image oriented clustering scheme. Although the clustering-based methods are

simple, the high iterate rate of each point in the point cloud leads to a high computation burden and defeats efficiency.

### 1.1.4. Learning Based Method

While deep learning-based methods often provide interesting results, the understanding of the type of coding solutions is essential to improve their design in order to be used effectively. To alleviate the cost of collecting and annotating large-scale point cloud data, Zhang and Zhu [31] propose an unsupervised learning approach to learn features from an unlabeled point cloud dataset by using part contrasting and object clustering with deep graph convolutional neural networks (GCNNs). Xu et al. [32] incorpate clustering method with the proposed FPCC-Net specially for industial bin-picking. *PointNet* [33] is the first machine learning neural network to deal with 3D raw points, and senstive to each points' immediate structure and geometry. Other current methods use deep learning directly on point clouds [34–37] or projections into a camera image [38] to segment instances in point clouds. Learning-based methods provide in an indoor scene but commonly suffer from long runtime and process large-scale point clouds.

### 1.2. Motivations

Although classical segmentation approaches mentioned above achieve promising segmentation results, one main drawback is huge computation-consuming, restricting their application to real-world, large-scale point cloud processing. To overcome this deficiency, we summarized two main strategies existing to accelerate the point cloud segmentation:

### 1.2.1. GPU vs. CPU

Conventional segmentation methods depend on the CPU's processing speed to run computations in a sequential fashion. Buys and Rusu [39] provide GPU-based version of EC [12] in the PCL [40] library and further extended by Nguyen et al. [41] who achieve 10 times speedup than the CPU-based EC.

Despite the fact that GPU enables faster segmentation, it is not practical for hardware devices with limited memory capacity and computation resources, such as mobile phones and small robotic systems (e.g., UAV), not to mention the steep price.

### 1.2.2. Pre-Knowledge vs. Unorganized

Taking advantage of pre-knowledge about the point cloud such as layer-based organized indexing in LiDAR data [42], relative pose between the LiDAR sensor, or point cloud in each 3D scan(frame) [43] accelerate segmentation speed. These assumptions provided by the structured point cloud hold in specific scenarios such as autonomous vehicle driving.

However, these premises are not available in many applications since not all the point cloud data are generated from the vehicle-installed LiDAR. For example, airborne laser scanning, RGB-D sensors, and image-based 3D reconstruction supply general organized data instead, making the pre-knowledge approaches [42,43] fail.

From the discussion above, we found out that an *efficient* and *low-cost* solution to *general* point cloud segmentation is vital for real-world applications but absent from research literature.

### 1.3. Contributions

As shown in Table 1, we attack the general point cloud segmentation problem and place an emphasis on computational speed as compared to the works that are considered state-of-the-art. The process of segmentation is proposed to be completed in two parts by our approach: (i) ground points removal and (ii) the clustering of the remaining points into meaningful sets. The proposed solution underwent extensively rigorous testing on both synthetic and real data. The results provided conclusive evidence that our method is superior to the other existing approaches. A fast segmentation redirects precious hardware

resources to more computationally demanding processes in many application pipelines. The following is a condensed summary of the contributions that this work makes:

- We present a new Euclidean clustering algorithm to the point could instance segmentation problem by using point-wise against the cluster-wise scheme applied in existing works.
- The proposed fast Euclidean clustering (FEC) algorithm can handle the general point cloud as input without relying on the pre-knowledge or GPU assistance.
- Extensive synthetic and real data experiments demonstrate that the proposed solution achieves similar instance segmentation quality but $100\times$ speedup over state-of-the-art approaches. The source code (implemented in C++, Matlab, and Python) is publicly available on https://github.com/feihuzhang/LiDARSeg.

**Table 1.** Summary of the related works.

| Ref | [37] | [44] | [42] | [43] | [39] | [35] | [34] | [12] | [10] | [16] | [18] | [18] | FEC (Ours) |
|---|---|---|---|---|---|---|---|---|---|---|---|---|---|
| GPU-free | | ✓ | ✓ | | | | | ✓ | ✓ | ✓ | ✓ | ✓ | ✓ |
| Unorganized point cloud | | | | ✓ | | ✓ | ✓ | ✓ | ✓ | ✓ | ✓ | ✓ | ✓ |

## 2. Materials and Methods

Our method concludes two steps: (i) ground points removal and (ii) the clustering of the remaining points.

### 2.1. Ground Surface Removal

Cloud points on the ground constitute the majority of input data and reduce the computation speed. Besides, the ground surface affects segmentation quality since it changes input connectivity. Therefore, it is essential to remove the ground surface as a pre-processing. Many ground points extraction methods such as grid-based [45] and plane fitting [44] have been used in existing works. The cloth simulation filter (CSF) [46], which is robust on complicated terrain surfaces, was the one that we decided to utilize to extract and eliminate ground points for this research.

### 2.2. Fast Euclidean Clustering

Similar to EC [12], we employ Euclidean (L2) distance metrics to measure the proximity of unorganized points and aggregate commonalities into the same cluster, which can describe as:

$$\min\|\mathbf{P}_i - \mathbf{P}_{i'}\|_2 \geqslant d_{\mathbf{th}} \tag{1}$$

where $\mathbf{C}_i = \{\mathbf{P}_i \in \mathbf{P}\}$ is a distinct cluster from $\mathbf{C}'_i = \{\mathbf{P}'_i \in \mathbf{P}\}$, and $d_{\mathbf{th}}$ is a maximum distance threshold.

Algorithm 1 describes the algorithmic processes and illustrates them with an example displayed in Figure 2. Note that the proposed algorithm uses **point-wise scheme**, which loop points with the input numbering order against the **cluster-wise scheme** used in EC and RG. The deployment of the proposed FEC is simple, requiring only 40 lines of code written in C++.

*Time complexity.* The complexity of constructing the kd-tree in the Big-O notation format is $O(3N \log N)$, where $N$ is the input size of the total point number in $\{\mathbf{P}\}$. The cost of main loop is $O(N^2 v)$, where $v$ is an constant number determined by the 3D point density $\rho$ and the neighbor radius threshold $d_{\mathbf{th}}$ as $v = \frac{4}{3}\pi d_{\mathbf{th}}{}^3 \rho$. Since $N^2 v > 3n \log N$, thus the overall cost is $O(N^2 v)$.

**Algorithm 1:** PROPOSED FEC

**Input:**
- $\mathbf{p}_i \in \mathbf{P}$: unorganized point cloud
- $d_{\mathbf{th}}$: the neighbor radius threshold
- $Th_{\max}$: the max number of neighbor points

**Output:**
- $\mathbf{P}.lab$: point cloud data with labels

```
// Initialization:
```
$\mathbf{P}.lab \leftarrow 0$: initialize all point labels as 0;
$segLab \leftarrow 1$: Segment label ;
Constructing kd-tree for $P$;
```
// Main loop:
```
**foreach** $\mathbf{p}_i.lab$ *in* $\mathbf{P}$ **do**
    **if** $\mathbf{p}_i.lable = 0$ **then**
        $\mathbf{P}_{NN} = \mathbf{FindNeighbor}(\mathbf{p}_i, d_{\mathbf{th}})$;
        **if** $\exists Nonzero(\mathbf{P}_{NN}.lab)$ **then**
            $minSegLab = \mathbf{min}(\mathbf{Nonzero}(\mathbf{P}_{NN}.lab), SegLab)$;
        **else**
            $minSegLab = SegLab$;
        **end**
```
        // Segment merge loop:
```
        **foreach** $\mathbf{p}_j$ *in* $\mathbf{P}_{NN}$ **do**
            **if** $\mathbf{p}_j.lab > minSegLab$ **then**
                **foreach** $\mathbf{p}_k.lab$ *in* $\mathbf{P}$ **do**
                    **if** $\mathbf{p}_k.lab = \mathbf{p}_j.lab$ **then**
                        $\mathbf{p}_k.lab = minSegLab$;
                **end**
            **end**
        **end**
    **end**
    $SegLab + +$;
**end**
**return** *point labels*

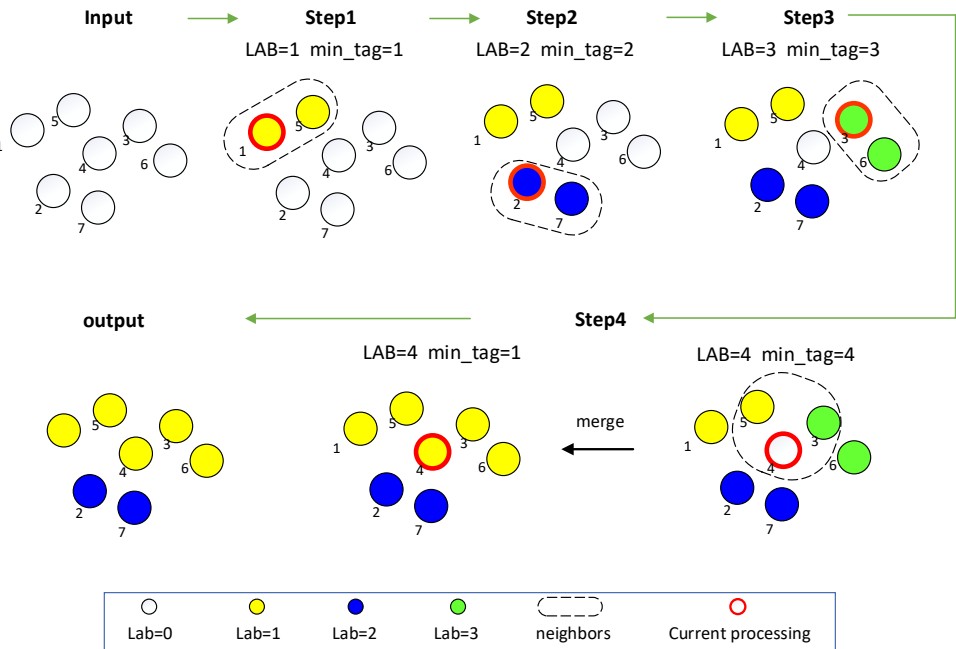

**Figure 2.** An example of FEC to point cloud segmentation. Note that FEC utilizes point-wise scheme with point index order.

### 2.3. Efficiency Analysis

The running time of the proposed FEC algorithm is analyzed in comparison to two different segmentation methodologies (i.e., RG [10] and EC [12]) on several typical examples in order to gain an understanding of the algorithm's effectiveness. This result intuitively explains the observations in our experiments that the FEC has significant advantages in terms of efficiency against others.

***Setting.*** We assume a point cloud $\{\mathbf{P}\}$ with $n$ clusters $\{\mathbf{C}_l \mid l \in [1, n]\}$. The number of points in $\{\mathbf{C}_i\}$ is $m_i$. Without losing the generality to investigate the limits of running time of EC, RG, and FEC, we draw three extreme distribution cases as shown in Figure 3. $k_n$ represents the number of operations on all points in the inner loop of the algorithm, which is affected by the size of the point cloud. $k_{sc}$ represents the number of searches for the kd-tree, which is affected by the number of clusters. Without loss of generality, we verify the value ranges of $k$ in three extreme situations: Case 1, all points are merged into one cluster; Case 2, each point belongs to a cluster; Case 3, all points are distributed on a line. In all three cases, the value of $k_{sc}$ is significantly less than $k_n$. While according to the Equation (3), the running time of RG and EC is positively correlated with $k_n$, and the running time of FEC is only affected by $k_{sc}$. Note that all three algorithms use a triple nested loop. Thus, we denote the single kd-tree search $\Omega(.)$ takes $t_1$ time while the other $O(1)$ cost processes take $t_2$, such as value assignment. The function for counting numbers is denoted by $\pi(.)$. Since FEC generates clusters by merging sub-clusters, we assume $\Theta(\mathbf{C}_l)$ is the function to get the sub-cluster sets.

***Time consuming.*** With such a setting, the processing time of EC, RG, and FEC on $\{\mathbf{P}\}$ are:

$$\begin{aligned} t^{\mathrm{EC}} &= Nt_1 + k_{\mathrm{n}}t_2 \\ t^{\mathrm{RG}} &= Nt_1 + 2k_{\mathrm{n}}t_2 \quad \text{with,} \quad \begin{array}{l} 2N - 2n \leqslant k_{\mathrm{n}} < N^2 \\ 1 \leqslant k_{\mathrm{sc}} \leq N - n \end{array} \\ t^{\mathrm{FEC}} &= k_{\mathrm{sc}}t_1 + k_{\mathrm{sc}}Nt_2 \end{aligned} \tag{2}$$

where $k_{\mathrm{n}}$ and $k_{\mathrm{sc}}$ defined as:

$$k_{\mathrm{n}} = \sum_{l=1}^{n} \pi(\Omega(\sum_{\in \{\mathbf{C}_l\}} \mathbf{P}_i)), \quad k_{\mathrm{sc}} = \sum_{l=1}^{n} \pi(\Theta(\mathbf{C}_l)) \tag{3}$$

Note that EC and RG loop every point in $\mathbf{C}_i$ and thus constantly consume $Nt_1$. Alternatively, FEC calls the number of sub-clusters times $\Omega(.)$, depending on the numbering order.

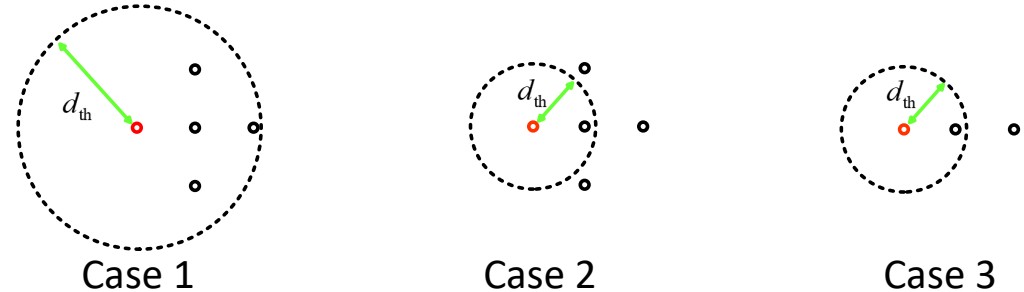

**Figure 3.** Values of $k_{\mathrm{n}}$ and $k_{\mathrm{sc}}$ in three extreme cluster distribution cases. Case 1 (maximus $d_{th}$): $k_{\mathrm{sc}} = 1$, $k_{\mathrm{n}} = N^2$; Case 2 (minimus $d_{th}$): $k_{\mathrm{sc}} = N - n$, $k_{\mathrm{n}} = 2N - 2n$; Case 3 (all points aligned in a line): $k_{\mathrm{sc}} = \frac{1}{2}(N - n)$, $k_{\mathrm{n}} = 2N - 2n$.

In the inner loops, EC and RG check if the neighbourhood points meets the criterion (minimum distance for EC, normal and curvature for RG) to be added into $\mathbf{C}_l$, and leads to $k_{\mathrm{n}}t_2$ and $2k_{\mathrm{n}}t_2$ respectively. While FEC needs to loop whole $\{\mathbf{P}\}$ with $Nt_2$ consuming in every sub-cluster merging.

To analyze the efficacy of the aforementioned three algorithms, we investigate the ratio between the total running time of the suggested FEC and that of the others:

$$\frac{t^{\text{FEC}}}{t^{\text{RG}}} < \frac{t^{\text{FEC}}}{t^{\text{EC}}} = \frac{k_{\text{sc}}(t_1 + Nt_2)}{Nt_1 + k_{\text{n}}t_2} < \frac{k_{\text{sc}}(t_1 + Nt_2)}{Nt_1 + Nt_2} < 1 \tag{4}$$

- In general setting, $k_{\text{sc}} \ll N$ which makes $\frac{t^{\text{FEC}}}{t^{\text{RG}}} < \frac{t^{\text{FEC}}}{t^{\text{EC}}} \approx \frac{k_{\text{sc}}}{N} \ll 1$ and lifting efficiency of FEC over EC and RG.
- By substituting the lower bound of $k_{\text{n}}$ and upper bound of $k_{\text{sc}}$ in Equation (2) into $t^{\text{EC}}$, $t^{\text{RG}}$ and $t^{\text{FEC}}$, FEC is faster than EC and RG even with the extreme setting.

We interpolate the improvement of FEC raised by the point-wise scheme against the cluster-wise scheme used in EC and RG. This difference leads to significantly fewer calls to kd-tree search $\Omega(.)$ in the first loop.

## 3. Experiments and Results

### 3.1. Method Comparison

In our experiments, the proposed method *FEC* was compared to five state-of-the-art point cloud segmentation solutions:

- *EC*: Classical Euclidean clustering algorithm [12] implemented in PCL library [40] (*EuclideanClusterExtraction* function).
- *RG*: Classical region growing based point cloud segmentation solution [10] implemented in PCL library [40] ( *RegionGrowing* function).
- *SPGN*: Recent learning-based method [34] which is designed for small-scale indoor scenes (https://github.com/laughtervv/SGPN, Salt Lake City, USA).
- *VoxelNet*: Recent learning-based method [35] which learns sparse point-wise features for voxels and use 3D convolutions for feature propagation (https://github.com/steph1793/Voxelnet, Salt Lake City, USA).
- *LiDARSeg*: State-of-the-art instance segmentation [37] for large-scale outdoor LiDAR point clouds (https://github.com/feihuzhang/LiDARSeg, Paris, France).

Note that the value of $d_{\text{th}}$ and $\text{Th}_{\text{max}}$ were set as the same to **EC**, and **RG**. For a fair comparison, we remove the ground points in the point cloud (Section 2.1) and then use them as the input for **EC**, **RG**, and **FEC**.

### 3.2. Metrics Evaluation

We provide three metrics (average prevision, time complexity, space complexity) to demonstrate that the proposed **FEC** outperforms baselines on efficiency without penalty to effectiveness.

- *Average precision:* The **average precision (AP)** is a widely accepted point-based metric to evaluate the segmentation quality [37], as well as similar to criterion for COCO instance segmentation challenges [47]. The equation for AP can be presented as:

$$AP = \frac{TP}{TP + FP} \tag{5}$$

  where TP is true positive, and FP is false positive. We use 0.75 as the point-wise threshold for true positives in the following experiments.
- *Complexity:* Both running time and memory consumption of real data experiments are designed to evaluate the time complexity and the space complexity. Our method was executed on a 32GB RAM, Intel Core I9 computer and compared to two other classical geometry-based methods: **EC** and **RG**. The NVIDIA 3090 GPU was utilized to evaluate both the learning-based methods **SPGN** and **VoxelNet**.

### 3.3. Synthetic Data

In this experiment, we evaluate the performances of **EC**, **RG**, and **FEC** respectively on synthetic point cloud data with increasing scale. Note that we generate unorganized points where the state-of-the-art learning-based methods **SPGN**, **VoxelNet**, and **LiDARSeg** which use single LiDAR scan as input, are not feasible in such setting.

### 3.3.1. Setting

We first divide 3D space into multiple voxels, and then generate clusters (segments) by filling *m* 3D points evenly inside the randomly selected *n* voxels. Under such a setting, we can control the cluster number by varying the value of *n*. Similarly, the cluster density can be determined by varying the value of *m*. Besides, we can also simulate clusters with different uniformities by changing the strategy of filling 3D points into voxel from even to random (even filling followed by random shift with variance $\sigma$). Note that we randomly set the 3D point index to make the synthetic an unorganized point cloud.

### 3.3.2. Varying Density of Cluster

In this experiment, we increase the density of synthetic data from 10 to 500 under varying total cluster numbers from 100 to 2200. As illustrated in Figure 4, the running time of both **RG** and **EC** grew significantly with increasing cluster densities under varying cluster numbers. An interesting observation is that the growth curve of **RG** is linear while **EC** provides exponential growth and overtakes the running time with a density of cluster larger than 500. In contrast, the proposed method **FEC** provides stable running times which are at least 100× faster performance over **EC** and **RG** with increasing density (number of points in unit volume) of each cluster.

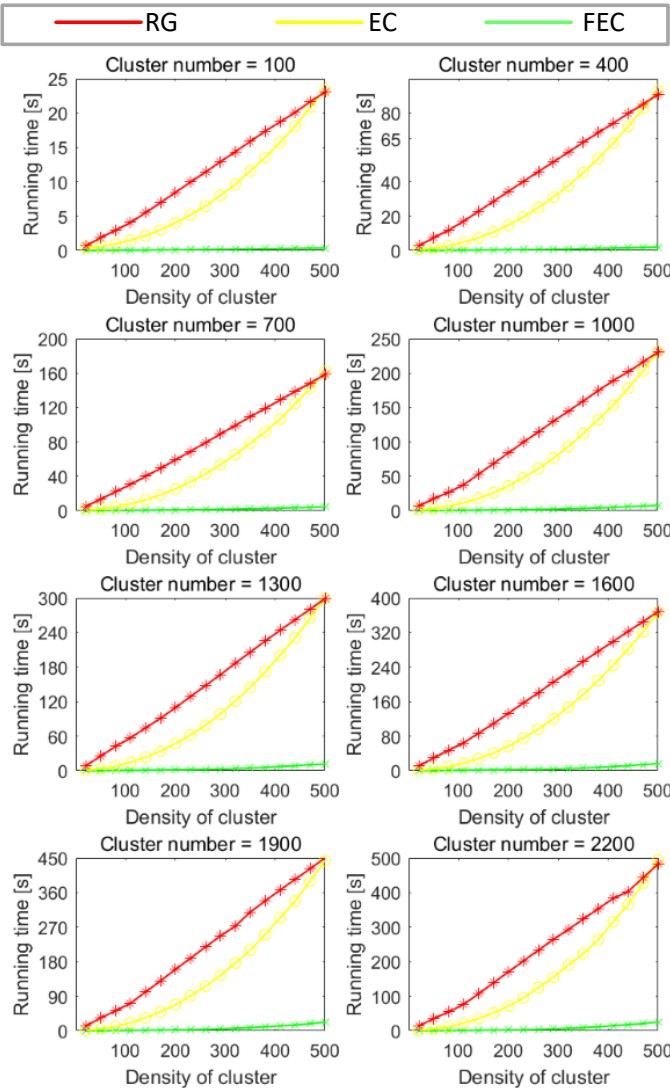

**Figure 4.** Running time for **EC** (yellow), **RG** (red) and **FEC** (green) with increasing density of different number of cluster.

### 3.3.3. Varying Number of Clusters

In this experiment, we fix the density of clusters to 200, and increase the number of clusters from 100 to 1900. Figure 5a demonstrates that the running time of **EC** and **RG** grow dramatically with an increasing number of clusters from milliseconds level to hundred seconds level. In contrast, **FEC** runs significantly faster with < 1 s performance under all configurations.

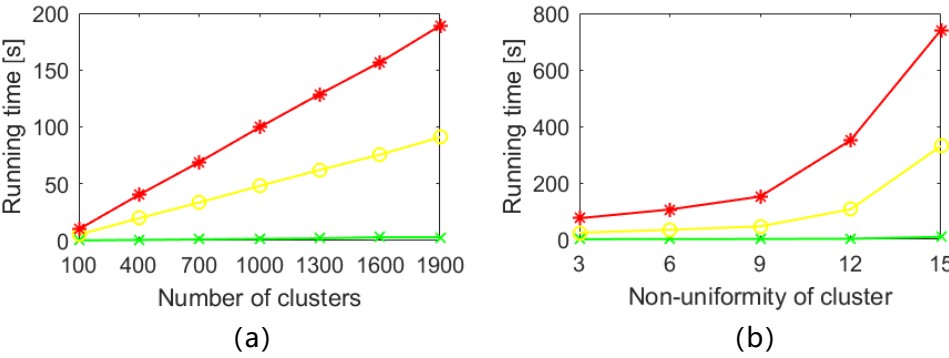

(a)                                                                                    (b)

**Figure 5.** Running time for **EC** (yellow), **RG** (red) and **FEC** (green) with increasing number of clusters (**a**), and non-uniformity (**b**) of each cluster.

### 3.3.4. Varying Uniformity of Cluster

We alter the uniformity of each cluster in this experiment by simulating an increasing number of normal distribution sub-clusters. The results in Figure 5b reveal that the uniformity of the cluster obviously drags the running time of **EC** and **RG** with a hundredfold increase. In contrast, the uniformity of the cluster has slightly affected **FEC** without significant growth in running time. Besides, as shown in Figure 6 that the proposed method achieve fastest and stable performance with the increasing number of cluster and cluster density jointly.

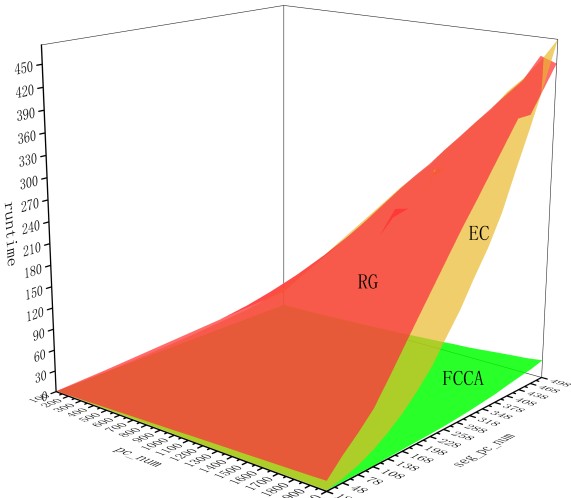

**Figure 6.** Running time for **EC** (yellow), **RG** (red) and **FEC** (green) with increasing dual-mixing of cluster number and density.

### 3.3.5. Segmentation Quality

In the synthetic data experiment, we observe that all three geometry-based methods **EC**, **RG**, and **FEC** provide segmentation precision approximate to 1 without significant difference.

*3.4. Real Data*

In this experiment, we evaluate the performance of all methods on a publicly available dataset, namely, KITTI odometry task [11] with point-wise instance label from semanticKITTI [48].

Following the instructions of [34,35,37], we trained the **SPGN**, **VoxelNet** and **LiDARSeg** using semanticKITTI. Thus, we compare the proposed method **FEC** against to classical geometry-based approaches **EC**, **RG** and learning based solutions **SPGN**, **VoxelNet**, **LiDARSeg** in 12 sequences of semanticKITTI.

3.4.1. FEC vs. Geometry-Based Methods on KITTI

We tested the **EC**, **RG** and the proposed **FEC** on real point cloud datasets from KITTI odometry point cloud sequence [11] with two common segmentation styles in practice, namely:

- **Inter-class segmentation:** the input for inter-class segmentation is a point cloud representing a single class, such as car, building, or tree, for example. Following the completion of the classification step, instance segmentation is carried out in a manner that is distinct for each of the classes.
- **Intra-class segmentation:** as input, intra-class segmentation utilizes multiple-class point clouds. In such a mode, the original LiDAR point cloud is utilized as input without classification.

*Efficiency.* As shown in Table 2 that quantitatively **FEC** achieves an average of 50× speedup against existing method **EC** and 100× **RG** under intra-class segmentation mode on 12 sequences form #03 to #11. In inter-class mode, **FEC** achieves average 30× and 40× speedup for car and building segmentation, 10× and 20× speedup for tree segmentation against to **EC** and **RG**. Besides, an interesting observation is that the running time of the three methods on #03 and #11 is nearly 2–3 times longer against the rest sequences under the intra-class mode. This is raised because the total number of instances in #03 and #11 is more extensive than the others, especially the tiny objects such as bicycles, trunks, fences, poles, and traffic signs. Since the geometry-based approaches will call more cluster processes in the loop with many small instances, thus, **RG**, **EC**, and **FEC** take much more running time on #03 and #11 sequences over the others.

**Table 2.** Experimental results on point clouds from KITTI vision benchmark [11]. Running time (in seconds *s*) of **EC**, **RG** and **FEC** on 11 sequences (#00-11 from odometry task) based on inter-class and intra-class styles are reported [in seconds]. Best results are shown in green.

| Dataset (Point Number) | Mode | Class (Point Number) | EC [12] (s) | RG [10] (s) | FEC [Ours] (s) |
|---|---|---|---|---|---|
| #00 (1,498,667) | intra-class | - | 414 | 780 | 140 |
| | inter-class | car (55,048) | 23.0 | 26.9 | 0.5 |
| | | building (131,351) | 13.0 | 58.1 | 1.3 |
| | | tree (351,878) | 51.9 | 167.0 | 9.9 |
| #01 (1,249,590) | intra-class | - | 289.4 | 651.0 | 103.3 |
| | inter-class | car (62,469) | 21.7 | 30.0 | 0.5 |
| | | building (204,911) | 26.3 | 95.9 | 2.2 |
| | | tree (455,519) | 150.3 | 232.7 | 6.3 |
| #02 (1,406,316) | intra-class | - | 323.2 | 721.3 | 105.9 |
| | inter-class | car (85,528) | 37.7 | 42.0 | 0.87 |
| | | building (209,704) | 21.1 | 98.3 | 2.4 |
| | | tree (476,077) | 118.1 | 242.2 | 23.3 |
| #03 (1,714,700) | intra-class | - | 931.1 | 908.1 | 90.4 |
| | inter-class | car (89,337) | 31.9 | 42.4 | 2.8 |
| | | building (210,300) | 21.4 | 97.8 | 2.39 |
| | | tree (582,929) | 167.1 | 298.6 | 17.2 |

**Table 2.** *Cont.*

| Dataset (Point Number) | Mode | Class (Point Number) | EC [12] (s) | RG [10] (s) | FEC [Ours] (s) |
|---|---|---|---|---|---|
| #04 (1,324,991) | intra-class | - | 313.6 | 682.6 | 82.1 |
| | inter-class | car (89,646) | 36.1 | 43.9 | 0.9 |
| | | building (212,195) | 98.2 | 99.3 | 2.33 |
| | | tree (653,342) | 292.4 | 335.9 | 18.0 |
| #05 (1,239,293) | intra-class | - | 319.3 | 631.3 | 44.0 |
| | inter-class | car (96,472) | 45.8 | 48.2 | 0.9 |
| | | building (241,977) | 173.6 | 122.7 | 2.6 |
| | | tree (658,529) | 127.3 | 335.1 | 21.0 |
| #06 (1,276,859) | intra-class | - | 319.3 | 631.3 | 44.0 |
| | inter-class | car (99,480) | 39.5 | 48.3 | 0.9 |
| | | building (241,977) | 36.0 | 124.0 | 3.1 |
| | | tree (679,665) | 158.9 | 345.8 | 25.5 |
| #07 (1,556,851) | intra-class | - | 413.1 | 801.3 | 123.5 |
| | inter-class | car (102,205) | 38.8 | 51.0 | 1.0 |
| | | building (274,741) | 195.1 | 134.6 | 3.4 |
| | | tree (751,354) | 156.6 | 383.3 | 32.2 |
| #08 (1,828,399) | intra-class | - | 265.7 | 417.3 | 49.1 |
| | inter-class | car (113,404) | 33.5 | 54.2 | 1.8 |
| | | building (307,844) | 55.4 | 156.8 | 3.9 |
| | | tree (754,161) | 189 | 395.9 | 26.3 |
| #09 (1,116,058) | intra-class | - | 225.7 | 563.5 | 62.7 |
| | inter-class | car (115,489) | 36.0 | 55.8 | 1.3 |
| | | building (296,751) | 55.9 | 153.53 | 3.8 |
| | | tree (828,874) | 229.3 | 428.6 | 35.1 |
| #10 (1,172,405) | intra-class | - | 280.4 | 598.5 | 59.9 |
| | inter-class | car (115,617) | 39.4 | 55.8 | 1.0 |
| | | building (311,047) | 43.0 | 152.9 | 3.9 |
| | | tree (841,600) | 230.4 | 431.2 | 24.8 |
| #11 (2,002,769) | intra-class | - | 948.2 | 1062.9 | 120.9 |
| | inter-class | car (127,417) | 51.7 | 61.6 | 1.3 |
| | | building (337,911) | 48.6 | 166.6 | 4.3 |
| | | tree (870,785) | 274.7 | 447.1 | 23.2 |

*Memory-consuming.* As shown in Table 3 that the proposed **FEC** consume only one third and half of memory against to **EC** and **RG**.

**Table 3.** Running time (in seconds), memory-consuming (in MB), and AP (average precision defined by [37]) over 11 sequences in KITTI [11] odometry task are reported. Best and second best results are shown in green and blue.

| Method | Time | | | | Memory (MB) | AP (%) |
|---|---|---|---|---|---|---|
| | Intra-Class | Inter-Class | | | | |
| | | Car | Building | Tree | | |
| EC | 387.8 | 36.3 | 65.7 | 195.6 | 211 | 65.8 |
| RG | 706.0 | 46.7 | 121.7 | 337.0 | 145 | 61.1 |
| FEC | 87.8 | 1.2 | 3.0 | 21.9 | 89 | 65.5 |

*Effectiveness.* As shown in Table 3 quantitatively, all three geometry-based methods provide similar instance segmentation quality with AP larger than 60%. Specifically, **EC** provides the best segmentation accuracy at 65.8% while the proposed **FEC** achieves a slightly lower score at 65.5%. The qualitative segmentation results are shown in Figure 7 with sequence #00 and #11 as two examples. It verifies our statement that our method and baseline methods provide similar segmentation quality globally. Interestingly, we found out that the proposed **FEC** provides better segmentation quality in handling details segment.

As shown in Figure 8, for the points on the tree, which are sparer and nonstructural than the other classes, EC and RG often suffer from over-segmentation and under-segmentation problems. For example, in the second row of Figure 8, the three independent trees are clustered into the building by EC and RG while FEC successfully detects them. In summary, as shown in Table 3 demonstrates that our method achieves significant improvement in efficiency while *without penalty to the performance (quality)*.

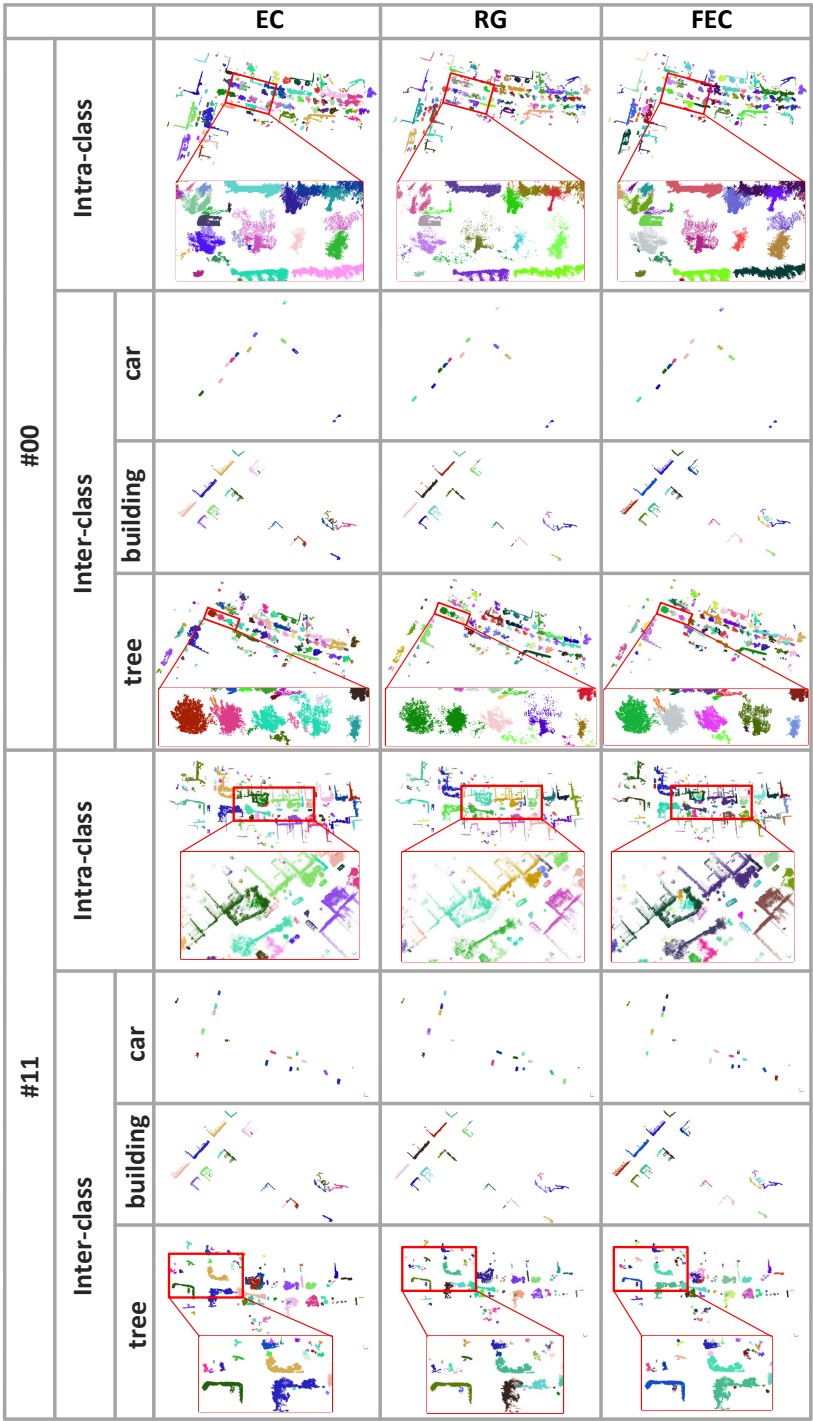

**Figure 7.** Qualitative segmentation results of *EC, RG* and *FEC* on KITTI odometry dataset [11] sequence #00 and #11.

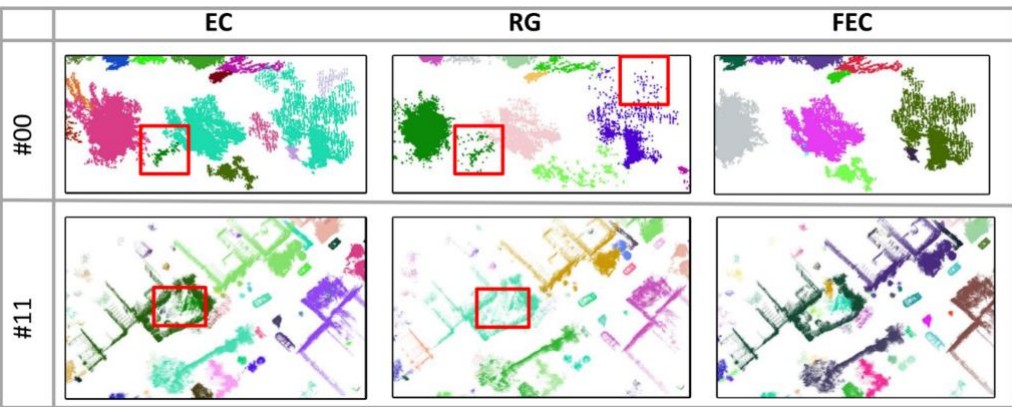

**Figure 8.** Details and highlights of segmentation results of *EC*, *RG* and *FEC* on KITTI odometry dataset [11] sequence #00 and #11.

### 3.4.2. FEC vs. Learning-Based Methods on KITTI

In this experiment, we compare the proposed method **FEC** to state-of-the-art learning-based solutions on sequence #00-#11 of the KITTI odometry dataset [48]. We trained the learning approaches **SPGN** [34], **VoxelNet** [35] and **LiDARSeg** [37] semanticKITTI labeling [48], and evaluated the running time and average precision (AP) defined in [37]. Note that all the learning-based methods were tested with GPU while the geometry solutions **EC** [12], **RG** [10], and **FEC** run with CPU only.

**Data input.** Note that in the Section 3.4.1 we use the point cloud of the whole sequence as input to three geometry-based methods. However, such large scale input is not feasible to the state-of-the-art deep learning based approaches for instance segmentation or 3D detection, namely, **SGPN**, **VoxelNet**, and **LiDARSeg**. Thus, in this experiment, we alternatively spilt the 12 sequences in the KITTI odometry dataset into a single scan as input. Besides, we also report the performances of **EC** and **RG**.

**Training and hardware.** Particularly, we use the ground-truth point-wise instance label from semanticKITTI panoptic segmentation dataset [48] to train textbfSGPN, **VoxelNet**, and **LiDARSeg** with a Nvidia 3090 GPU while forcing the CPU-only mode for **EC**, **RG** and proposed **FEC**.

**Efficiency.** Please note that since **LiDARSeg** [37] is designed for segmenting the single LiDAR scan, thus the running time we recorded below is the average process time of each scene instead of the whole sequence. As shown in Table 4 that **FEC** achieves 5× speed up against to **EC**, **VoxelNet** and **LiDARSeg**, 10× to **RG**, and 20× to **SPGN**. Since all the geometry-based methods require ground surface removal as pre-process, thus for a fair comparison, the time-consumings of ground surface detection and removal have been considered in the total running time. It is important to notice that **FEC** relies on CPU calculation only while all the learning-based approaches are accelerated with GPU in inference without mentioning the huge time-consuming in the training stage.

**Effectiveness.** As shown in Table 4 that quantitatively all three geometry-based methods **EC**, **RG** and **FEC** achieve similar instance segmentation quality with AP around 62%. Note that the segmentation accuracy of geometry-based approaches is even slightly better than the leaning-based ones with AP at 59% of LiDARSeg, 55.7% of **VoxelNet** and 46.3% of **SPGN**. The qualitative comparisons are shown in Figure 9 with 2 scans in sequence #00 as examples. We can observe that the proposed solution **FEC** method provides similar segmentation quality as the state-of-the-art learning-based method **LiDARSeg**.

**Table 4.** Comparisons to state-of-the-art learning based solutions sequence #00–#11 of KITTI odometry dataset [48]. Note that since **LiDARSeg** [37] is designed for segmenting the single LiDAR scan, thus the running time we reported is the average process time of each scene instead of the whole sequence. Best results are shown in green.

| Sequence | GPU: Nvidia 3090X1 | | | CPU: Intel I9 | | |
|---|---|---|---|---|---|---|
| | SPGN [34] (ms) | VoxelNet [35] (ms) | LiDARSeg [37] (ms) | FEC [Ours] (ms) | RG [10] (ms) | EC [12] (ms) |
| #00 | 1510.9 | 442.7 | 470.2 | 140.8 | 780.8 | 414.6 |
| #01 | 1705.4 | 454.1 | 471.9 | 103.3 | 651.0 | 289.4 |
| #02 | 1605.6 | 427.4 | 504.8 | 105.9 | 721.3 | 323.2 |
| #03 | 1875.3 | 380.6 | 386.0 | 90.4 | 908.1 | 931.1 |
| #04 | 1523.2 | 469.0 | 506.2 | 82.2 | 682.7 | 313.7 |
| #05 | 1675.0 | 453.4 | 383.1 | 44.0 | 631.4 | 319.3 |
| #06 | 1749.4 | 606.8 | 481.1 | 71.7 | 653.7 | 318.5 |
| #07 | 1496.2 | 460.1 | 439.9 | 53.6 | 801.4 | 123.6 |
| #08 | 1816.3 | 644.2 | 452.7 | 49.1 | 417.3 | 165.8 |
| #09 | 1509.2 | 472.7 | 465.9 | 62.8 | 563.5 | 225.8 |
| #10 | 1558.0 | 396.4 | 485.6 | 60.0 | 598.5 | 280.4 |
| #11 | 1485.6 | 362.9 | 422.0 | 121.0 | 1062.9 | 948.3 |
| Average time (ms) | 1625.8 | 464.2 | 455.8 | 87.9 | 706.0 | 387.8 |
| AP (%) | 46.3 | 55.7 | 59.0 | 63.2 | 61.0 | 63.1 |

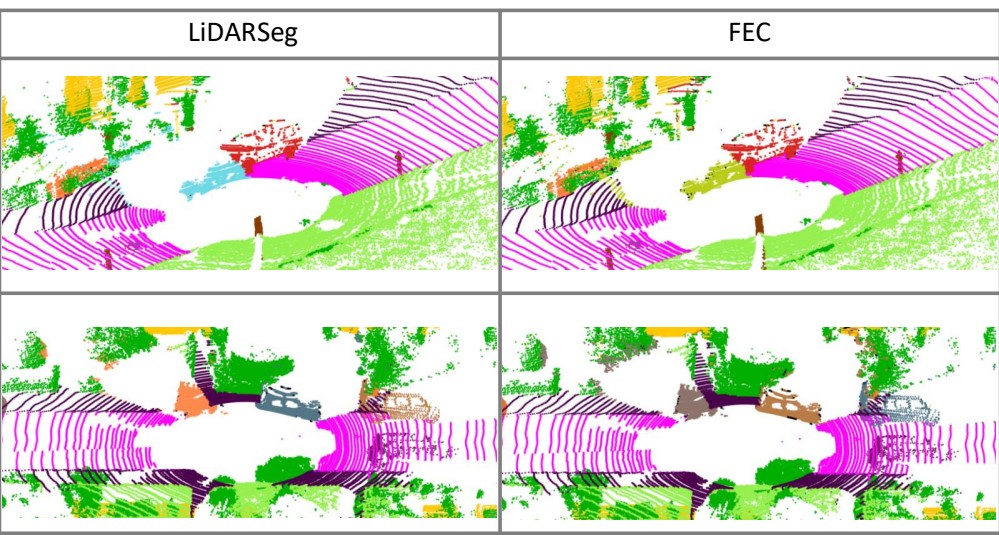

**Figure 9.** Comparisons to state-of-the-art learning based solution **LiDARSeg** [37] on KITTI odometry dataset [48] with semanticKITTI labeling [48]. Note that **LiDARSeg** [37] only applies to a single LiDAR scan as input instead of a long sequence.

## 4. Discussion

### *If FEC is faster?*

Both the synthetic (Section 3.3) and real data (Section 3.4) experiments demonstrate that the proposed **FEC** outperform the state-of-the-art geometry-based and learning-based methods with a significant margin. Specifically, our method is an order of magnitude faster than **RG** [10] and nearly 5 times faster than **EC** [12]. Besides, without GPU acceleration, **FEC** is nearly 5 times faster **LiDARSeg** [37], 6 times faster than **VoxelNet** [35], and nearly 20 times faster than **SPGN** [34].

### *If FEC losses accuracy?*

One may have concern that if **FEC** will lose segmentation accuracy in order to accelerate the processing speed. Both the synthetic (Section 3.3) and real data (Section 3.4) experiments verify that **FEC** provides similar segmentation quality as the conventional geometry-based methods **RG** [10], **EC** [12], and even slightly better than the learning-based solutions **LiDARSeg** [37], **VoxelNet** [35] and **SPGN** [34]. Thus, we point out that the proposed solution **FEC** achieves significant improvement in efficiency while ***without penalty to the performance (quality)***.

### *Why FEC is faster?*

Based on the intuitive analysis in the Section 2.3, we interpret the **FEC** brings significant improvement , n efficiency mainly due to the point-wise scheme over the cluster-wise scheme used in existing works **RG** [10] and **EC** [12]. Such a novel point-wise scheme leads to significantly fewer calls to kd-tree search in the loop, which is the key to reducing the running time significantly.

### *Where we can use FEC?*

The proposed solution **FEC** is a GPU-free faster point cloud instance segmentation solution. It can handle general point cloud data as input without relying on scene scale (e.g., single LiDAR scan) or structure pre-knowledge (e.g., scan line id). Thus, we can apply **FEC** to large-scale point cloud instance segmentation in various 3D perception (computer vision, remote sensing) and 3D reconstruction (autonomous driving, virtual reality) tasks.

## 5. Conclusions

This paper introduces an efficient solution to a general point cloud segmentation task based on a novel algorithm named faster Euclidean clustering. Our experiments have shown that our methods provide similar segmentation results but with 100× higher speed than the existing approaches. We interpret this improved efficiency as using the point-wise scheme against the cluster-wise scheme in existing works.

**Future work.** The current implementation of **FEC** is based on a serial computation strategy. Since the point-wise scheme contains multiple associations between the outer and inner loops, thus the parallel computation strategy could be applied to **FEC** for a potential acceleration.

**Author Contributions:** Conceptualization, Y.C., Y.X. and Y.L.; methodology, Y.C., Y.W. and Y.L.; validation, Y.C. and H.Z.; formal analysis, Y.C. and Y.L.; investigation, Y.W. and H.Z.; resources, Y.L.; data curation, Y.C. and H.Z.; writing—original draft preparation, Y.C., Y.X. and Y.L.; writing—review and editing, Y.X. and Y.L.; visualization, Y.C., Y.L.; supervision, Y.X. and Y.L.; project administration, Y.L.; funding acquisition, Y.X. and Y.L. All authors have read and agreed to the published version of the manuscript.

**Funding:** This research was funded by (1) Nature Science Foundation of China (Grant No. 62102145), (2) Jiangxi Provincial 03 Specific Projects and 5G Program (Grant No. 20212ABC03A09), (3) Scientific and Technological Innovation Project of Jiangxi Provincial Department of Natural Resources (Grant No. ZRKJ20222312).

**Institutional Review Board Statement:** Not applicable.

**Informed Consent Statement:** Not applicable.

**Data Availability Statement:** Not applicable.

**Conflicts of Interest:** The authors declare no conflict of interest.

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
