# Peer review of "FEC: Fast Euclidean Clustering for Point Cloud Segmentation"

_drones, doi:10.3390/drones6110325_

Round 1

Reviewer 1 Report

Summary:

Cao et al. proposed an innovative Euclidean cluster approach for offline point cloud segmentation. Instead of a conventional cluster-wise scheme, they alter to a point-wise method to improve computing efficiency. The results indicate that the proposed FEC significantly outperforms SOTAs without many device requirements.

Strengths:

- Clarity: This manuscript is easy to read and mostly friendly to follow. Figure descriptions are handy for readers to comprehend the authors' targets and focus.

- Methods: The authors express adequately and clearly the FEC description. Considering extreme situation are very helpful to a solid methodology intuition.

- Algorithm evaluation: It is impressive that the authors analyze the proposed algorithm's time and space complexity ahead of the experiment.

- Experiment: The authors showed their confidence on FEC when competing with several SOTAs. They made abundant comparisons on both real and synthetic data. Furthermore,  the comprehensive qualitative research on cluster number, density, and uniformity also reveals that the proposed one defeats conventional methods all around.

Question & weakness

I will not say that there are so-called weaknesses in the manuscript, but several minor improvements may be beneficial:

- I would like to recommend that the author add details on hypotheses in the Abstract section, which is helpful to glimpse reading.

- It will be more friendly for readers if add unit notations (e.g., s) in the table head for long tables like Table 2 & 4.

Reviewer 2 Report

The paper presents the euclidean clustering algorithm using a point-wise against the cluster-wise scheme and the authors claim the 100x speed as compared to the previous findings. The paper is very well written, but there is a need to emphasize the contributions in a discrete manner.  What is the reason that your proposed method will give a low complexity as compared to the existing algorithms? I think the Table 4 caption needs to be modified and all the explanations which are in the caption will be included in the text. 

In Table 2, for #11 (2,002,769), why was the simulation time more as compared to the others, and for #03 (1,714,700), why was the simulation time too much less as compared with the others? Please elaborate comprehensively. 

Line 295 "Verifies our statement that our method and baseline methods provide similar segmentation quality globally. Interestingly, we found out that the proposed FEC provides better segmentation quality in handling details segment". There is a need to more explain the figure 7. 

Reviewer 3 Report

In this paper , the authors propose an efficient method to segment 3D point clouds. The proposed method is of practical use especially for online-data processing with drones / robots. The English writting of this paper is moderate.  There are following minor issues to be justfied.

1.  All the convential geometry-based approaches, including the proposed FEC, require ground surface removal as a pre-process. However, the learning-based mentioned in the manuscript does not rely on the such step. Thus, is the running time comparison reported in section 3.4.2 and table 4 fair enough? If the authors have already taken the time consuming of such pre-process into the total running time, they should describe the timing method for clearance.

2. Had any multithreading or parallel computation strategies been applied to the proposed algorithm? From my point of view, since the main structure of FEC is three sequential loops, thus it is possible to parallelize it by taking advantage of widespread multicore and multithreaded processors. 

3. The reference format should be in accordance with the mdpi style.
